# Versatile Multi-stage Graph Neural Network for Circuit Representation

**Shuwen Yang**
School of Intelligence Science and Technology
Peking University
swyang@pku.edu.cn

**Zhihao Yang**
School of Software and Microelectronics
Peking University
zhihaoyang@stu.pku.edu.cn

**Dong Li**
Huawei Noah's Ark Lab
lidong106@huawei.com

**Yingxue Zhang**
Huawei Noah's Ark Lab
yingxue.zhang@huawei.com

**Zhanguang Zhang**
Huawei Noah's Ark Lab
zhanguang.zhang@huawei.com

**Guojie Song**[*]
School of Intelligence Science and Technology
Peking University
gjsong@pku.edu.cn

**Jianye HAO**[*]
Huawei Noah's Ark Lab
haojianye@huawei.com

## Abstract

Due to the rapid growth in the scale of circuits and the desire for knowledge transfer from old designs to new ones, deep learning technologies have been widely exploited in Electronic Design Automation (EDA) to assist circuit design. In chip design cycles, we might encounter heterogeneous and diverse information sources, including the two most informative ones: the netlist and the design layout. However, handling each information source independently is sub-optimal. In this paper, we propose a novel way to integrate the multiple information sources under a unified heterogeneous graph named **Circuit Graph**, where topological and geometrical information is well integrated. Then, we propose **Circuit GNN** to fully utilize the features of vertices, edges as well as heterogeneous information during the message passing process. It is the first attempt to design a versatile circuit representation that is compatible across multiple EDA tasks and stages. Experiments on the two most representative prediction tasks in EDA show that our solution reaches state-of-the-art performance in both logic synthesis and global placement chip design stages. Besides, it achieves a 10x speed-up on congestion prediction compared to the state-of-the-art model.

## 1 Introduction

Integrated circuits (ICs) are extensively used in modern electronic products like computers, smartphones, and cars. Electronic Design Automation (EDA) includes a set of tools for circuit design in different development stages especially **logic synthesis** stage and **placement** stage (Fig.1). As

---

[*]Corresponding Author

36th Conference on Neural Information Processing Systems (NeurIPS 2022).

the scale and complexity of circuits continuously grow, the design efficiency and precision of EDA tools have become an essential problem, which attracts researchers to adopt deep learning techniques to assist the circuit design process [1]. Remarkable progress has been made in predicting circuits' quality and practicability in the earlier stage of the chip design to speed up optimization and reduce design cost [2, 3]. For example, predicting congestion for circuits in physical design stage can help detect their flaws and avoid producing defective chips, and chip design production cycle time can be further saved if such prediction could be done in logic synthesis stage.

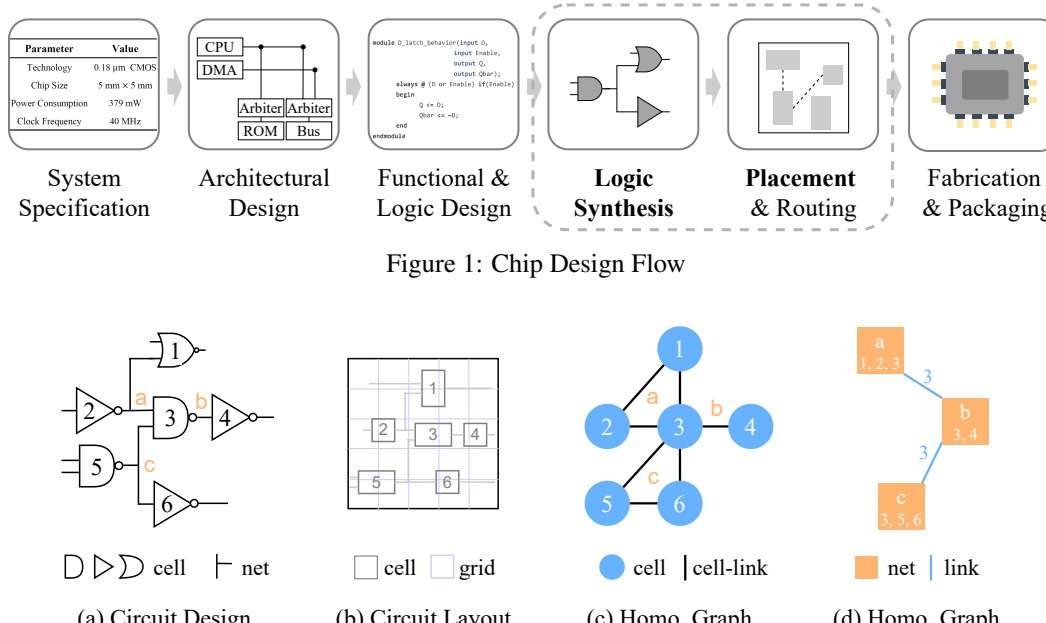

Figure 1: Chip Design Flow

Figure 2: Convert circuit design to a grid feature matrix and a homogeneous graph.

Because of the sophisticated process of circuit design, we will encounter diverse information sources from various design stages, among which logic synthesis and placement & routing determine the quality, e.g. time delay, of circuit and consume the majority of design pipeline time, as shown in Fig.2. In the logic synthesis stage, a circuit design is represented as a **Netlist** composed of *cells* and *nets*, where *cells* refer to the electronic units and *nets* refer to the connectivity (i.e. hyper-edges) among the *cells*. After the placement & routing stage, we will obtain the circuit layout with the position of each electronic unit provided after they have been fully placed on the circuit board. Besides, the downstream tasks might require learning representations on both *cells* and *nets*, tailoring for specific downstream tasks. Thus, how to better organize and fuse the diverse and heterogeneous information provided in the context of circuit representation is an important research question.

Although the information underlying the circuit design varies according to the input stages, we can divide it into **geometrical information** and **topological information**. The netlists only contain the topological information (e.g. *cell* type and logical relationship between *cells* and *nets*), while some geometrical information (e.g. positions of standard *cells*) is available after placement stage. According to which information they focus on, we categorize the existing prediction methods in circuit designs into **topological methods** and **geometrical methods**. The topological methods convert circuit designs into homogeneous graphs (Fig.2(c)(d)) and solve the problems with Graph Neural Network (GNN) [4, 5, 6], while most recent geometrical models convert the circuit designs into grid feature matrices (see Def.1 and Fig.2(b)) and adopt Computer Vision (CV) technologies to predict their properties [7, 8, 9].

However, the topological methods only consider the topological information in netlists and cannot effectively perceive geometrical structure introduced after the placement stage, so their performance on circuits after placement is greatly stifled. Besides, the geometrical models heavily rely on geometrical information and neglect the topology underlying the netlists, so they cannot handle circuits in stages earlier than global placement where geometry is not available. In a word, they can

only work well on circuits either in logic synthesis stage or placement stage but are not compatible with both. Moreover, most of the existing prediction methods design modules with prior knowledge to serve certain applications, so they are not flexible enough to handle diverse EDA tasks. Therefore, it is necessary to: (1) design a data structure to represent the circuit, which is adaptive to circuits in both logic synthesis and placement stages; (2) propose an efficient and effective circuit representation model which is compatible with two stages and various downstream tasks.

In this paper, we first convert the circuit design to **Circuit Graph** (see (Fig.3)), a heterogeneous graph which preserves most of the topological and geometrical information, with a linear time consumption to the scale of the design (see Appendix B.2 for proof). A circuit graph contains two types of vertices i.e. *cells* and *nets*. We use *pins* which connect *cells* and *nets* to represent the topology, which are also noted as *topo-edges*. When handling circuits in placement stage, we additionally link the *cells* which are geometrically close (named *geom-edges*) to represent the geometry.

Then, we design **Circuit GNN** to process Circuit Graphs with optimal efficiency and produce representations for *cells* and *nets* to handle diverse tasks on circuits in both logic synthesis stage and placement stage. To collect and enrich the topological and geometrical information, we conduct message-passing [10] on *topo-edges* and *geom-edges* individually and later fuse the messages to update *cells* and *nets* representations. Multiple layers of such "message-passing & fusion" inference further exploit the deep relationships between topology and geometry and facilitate Circuit GNN to output nutritious representations. Circuit GNN is also compatible with Circuit Graphs converted from circuits in logic synthesis stage because *geom-edges*'s absence will not disable the message-passing over *topo-edges* and the topological information can still be collected.

In summary, our contributions are:

- To address the challenge posed by the diverse and heterogeneous information source in the context of circuit representation, we propose a novel way to integrate the information named **Circuit Graph**, a heterogeneous graph where topological and geometrical information are integrated jointly, which is able to handle diverse circuit tasks on cell, nets level and on different stages. To our best knowledge, this is the first unified circuit representation approach that can be easily compatible across EDA tasks and stages.

- We propose a novel message-passing paradigm **Circuit GNN** that tailors the aforementioned graph dataset structure. We design message-passing on both topological and geometrical edges distinctively and then fuse the messages to update *cells* and *nets* representations. The efficiency and effectiveness of our design are demonstrated both methodologically and experimentally.

- Experiment results validate the superior performance and efficiency of Circuit Graph across multi-stages/tasks. For circuit congestion prediction task at logic synthesis stage, it improves the average grid-level accuracy by 16.7% against SOTA. At placement stage, it achieves 5.6% accuracy gain with 10x speed-up in congestion prediction task and 16.9% error gain in net wirelength prediction task.

## 2  Related Work

### 2.1  Topological Methods

The topological methods in EDA focus on the logic relationships between the *cells* and *nets* and usually reconstruct the circuit designs into graphs with vertices and edges. CongestionNet[5] and solutions in [6] link the *cell*-pairs connected via *nets* (Fig.2(c)) and adopt popular GNNs (e.g. GAT[11]) to generate *cell* representations for congestion prediction. To handle net length identification and net delay prediction, Net$^2$[12] links the *nets* connecting to one *cell* (Fig.2(d)) and designs a customized GNN to obtain *net* representations. All these methods perceive the circuit designs as homogeneous graphs and neglect the underlying heterogeneity, e.g. the interaction between *cells* and *nets* like signal input/output, so their solutions suffer from information loss and will affect the performance of downstream GNN.

## 2.2 Geometrical Methods

The geometrical methods in EDA focus on the spatial information of the circuit designs. A universal approach is to cut a circuit into small rectangles i.e. grids and convert it into RGB channels, where the grids are treated as pixels [8, 13]. Then, they encode the netlist structure and circuit features into green and blue channels while red channels are left for prediction targets (e.g. congestion). Finally, they use image translation methods (e.g. pix2pix [9]) to output new images with red channels filled and indirectly solve the EDA tasks.

The cutting-edge LHNN [14], however, converts the circuits into lattice networks [15] instead of images, where each grid serves as an internal node in the network and each net, as an external node, is connected to the grids it covers geometrically. LHNN successfully enhances topological information in geometrical method and achieves SOTA performance, but it can only handle congestion prediction task and can only work on circuits with placement information.

## 3 Circuit Graph

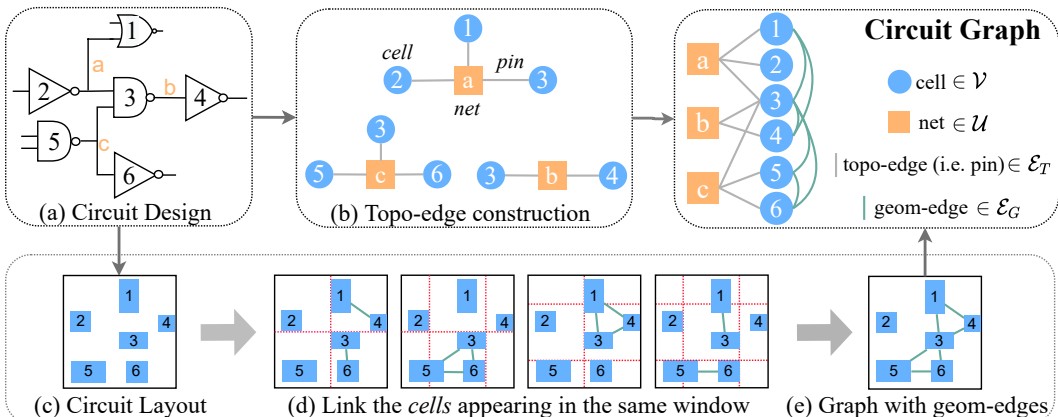

Figure 3: Convert circuit design to Circuit Graph. (b): Take *cell-net* connections i.e. *pins* as *topo-edges*. (c)(d)(e): Link the geometrically close *cell*-pairs to construct *geom-edges*.

### 3.1 Circuit Featurization

A circuit design is initially represented as a netlist composed of *cells* $\mathcal{V}$ and *nets* $\mathcal{U}$ (Fig.2(a)), and we define $\boldsymbol{X}_\mathcal{V}, \boldsymbol{X}_\mathcal{U}$ to be their feature matrices. $\boldsymbol{X}_\mathcal{V}$ mainly contains the size and degree (to *net*) of the cells, and $\boldsymbol{X}_\mathcal{U}$ stores the net span [14] and the degree (to *cell*).

Besides the basic attributes of *cells* and *nets*, the topology and geometry also play important roles in featurizing the circuit designs. The *pins* $\mathcal{P} \subseteq \mathcal{V} \times \mathcal{U}$ stand for the bipartite topology between *cells* $\mathcal{V}$ and *nets* $\mathcal{U}$, and $\boldsymbol{X}_\mathcal{P}$, the feature matrix of $\mathcal{P}$, preserves their interaction details e.g. the signal direction (input/output). After placement, the positions of *cells* $\boldsymbol{p}_x, \boldsymbol{p}_y$ are obtained, which serve as a major geometrical information. When carrying out deep learning on EDA, how to arrange the features of circuit design varies according the downstream model.

The geometrical methods cut the circuit into smaller rectangles i.e. *grids*[2] and generate raw features on them by synthesizing the positions of *cells* $\boldsymbol{p}_x, \boldsymbol{p}_y$ along with *cells*' and *nets*' features $\boldsymbol{X}_\mathcal{V}, \boldsymbol{X}_\mathcal{U}$ and their connections $\mathcal{P}$ (Fig.2(b)) [9, 14]:

**Definition 1** (Geometry-driven Circuit Featurization). $\mathcal{F}_G = \{\boldsymbol{X}_{gr}\}$, where $\boldsymbol{X}_{gr} = \mathbb{R}^{C_x \times C_y \times D_{gr}}$ is the feature matrix of grids. Note that $C_x, C_y$ are the column and row numbers of grids and $D_{gr}$ is the dimension of grids' raw features.

The *grid* feature $\boldsymbol{X}_{gr}$ mainly includes pin density and net density [7]. As the structure of $\boldsymbol{X}_{gr}$ is similar to RGB channels of image data, CV models (e.g. CNN[9]) can be easily adopted to handle EDA tasks where geometrical information are provided as input.

---

[2]In this paper, we use *grid* size $32\mu m \times 40\mu m$.

The topological methods prefer to construct the circuit design as a bipartite graph (Def.2) [12]:

**Definition 2** (Topology-driven Circuit Featurization). $\mathcal{F}_T = \{\mathcal{V}, \mathcal{U}, \mathcal{P}, \boldsymbol{X}_\mathcal{V}, \boldsymbol{X}_\mathcal{U}, \boldsymbol{X}_\mathcal{P}\}$, *where cells* $\mathcal{V}$ *and nets* $\mathcal{U}$ *are two types of vertices, and pins* $\mathcal{P}$ *are the edges connecting them.*

However, $\mathcal{F}_T$ is usually simplified as a homogeneous graph, and is fed to some popular GNNs, such as GAT (see Fig.2(c)(d) and Sec.2.1) [5, 6, 12], which leads to loss of heterogeneous information.

Note that the explicit raw features of *grid*, *cell*, *net* and *pin* are listed in Appendix A.

### 3.2 Definition of Circuit Graph

To better benefit from both the above two commonly used featurization system for circuit, we propose a novel way to encode both the topological and geometrical information (when available) underlying the circuits into a unified heterogeneous graph. As shown in Fig.3, we first take *pins* as *topo-edges* ($\mathcal{E}_T = \mathcal{P}$ and $\boldsymbol{X}_{\mathcal{E}_T} = \boldsymbol{X}_\mathcal{P}$). Then we link the geometrically-close *cells* with *geom-edges* $\mathcal{E}_G$ and store the *cell*-pair distances in feature matrix $\boldsymbol{X}_{\mathcal{E}_G}$. Finally, we define Circuit Graph as follows:

**Definition 3** (Circuit Graph). *A Circuit Graph* $\mathcal{G} = \{\mathcal{V}, \mathcal{U}, \mathcal{E}_T, \mathcal{E}_G, \boldsymbol{X}_\mathcal{V}, \boldsymbol{X}_\mathcal{U}, \boldsymbol{X}_{\mathcal{E}_T}, \boldsymbol{X}_{\mathcal{E}_G}\}$, *where* $\mathcal{V}, \mathcal{U}, \mathcal{E}_T \subseteq \mathcal{V} \times \mathcal{U}, \mathcal{E}_G \subseteq \mathcal{V} \times \mathcal{V}$ *refer to the set of cells, nets, topo-edges and geom-edges, respectively, and* $\boldsymbol{X}_\mathcal{V}, \boldsymbol{X}_\mathcal{U}, \boldsymbol{X}_{\mathcal{E}_T}, \boldsymbol{X}_{\mathcal{E}_G}$ *are their feature matrices.*

To preserve the topological information, we completely inherit $\mathcal{V}, \mathcal{U}, \mathcal{P}$ and their features from $\mathcal{F}_T$ in Def.2 rather than simplify them as homogeneous graphs with loss of heterogeneity. For geometrical information, to avoid calculating all $O(|\mathcal{V}|^2)$ *cell*-pairs' distances and reduce time cost to $O(|\mathcal{V}|)$, we split the *cells* by shifted windows[16] with size $(w_x, w_y)$ and link the *cells* with up to $c$ (named "link capacity") neighbouring *cells* located in the same window (see explicit steps in Appendix B.1). If we need to handle the problem at the pre-placement phase such as the logic synthesis stage, we will not have the *geom-edges* among the *cells* and will set $\mathcal{E}_G = \emptyset$ for the representation learning phase, which shows that Circuit Graph is compatible with circuits in logic synthesis stage.

Note that the time consumption of converting a circuit design into a Circuit Graph is $O(|\mathcal{V}|+|\mathcal{U}|+|\mathcal{P}|)$, which is linear to the scale of the circuit (see proof in Appendix B.2).

## 4 Circuit GNN

### 4.1 Overview

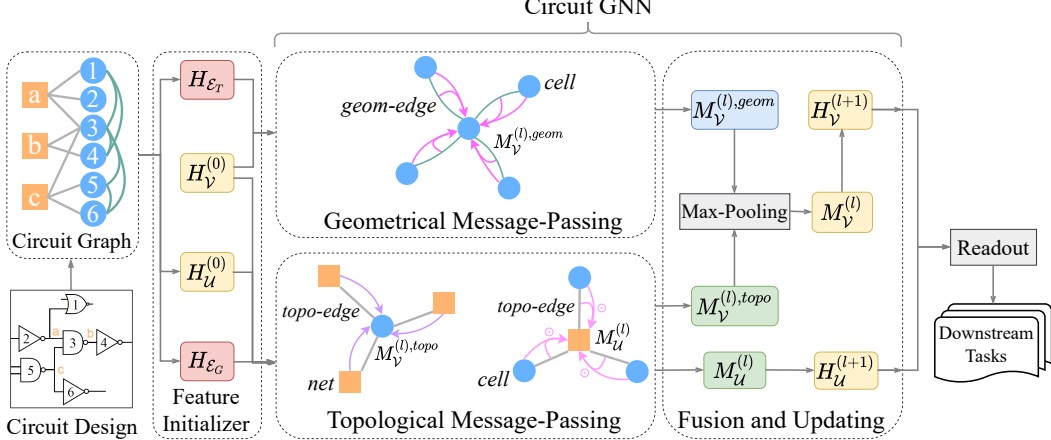

Figure 4: Framework of Circuit GNN. The topological messages between *cell* and *net* are passed through *topo-edge*, while the geometrical messages among *cells* are passed through *geom-edge*. Circuit GNN can be set to $L$ layers.

The framework of Circuit GNN is shown in Fig.4. We first input the Circuit Graph $\mathcal{G}$ and initialize the feature of *cells* $\mathcal{V}$, *nets* $\mathcal{U}$, *topo-edges* $\mathcal{E}_T$ and *geom-edges* $\mathcal{E}_G$ to hidden representations

$\boldsymbol{H}_{\mathcal{V}}^{(0)}, \boldsymbol{H}_{\mathcal{U}}^{(0)}, \boldsymbol{H}_{\mathcal{E}_T}, \boldsymbol{H}_{\mathcal{E}_G}$ with Multi-Layer Perceptrons (MLPs). Then deeper representations of *cells* and *nets* are generated via $L$ layers of circuit message-passing. Finally, the output *cell* and *net* representations are used for downstream tasks after passing through task-adaptive readout layers. Note that Circuit GNN's model sensitivity is justified in Appendix D.

## 4.2 Topo-Geom Message-passing

Inside each layer out of $L$ Topo-Geom Message-passing layers, topological and geometrical information are collected by passing messages through two types of heterogeneous edges ($\mathcal{E}_T$ and $\mathcal{E}_G$). Then, the messages are used to update *cell* representations $\boldsymbol{H}_{\mathcal{V}}$ and *net* representations $\boldsymbol{H}_{\mathcal{U}}$. In Topological Message-passing, the messages between *cells* $\mathcal{V}$ and *nets* $\mathcal{U}$ are transmitted through *topo-edges* $\mathcal{E}_T$:

$$\boldsymbol{M}_{\mathcal{V}}^{(l),topo} = \Phi_{msg}^{\mathcal{U}\xrightarrow{\mathcal{E}_T}\mathcal{V}}(\mathcal{U}, \mathcal{E}_T, \boldsymbol{H}_{\mathcal{U}}^{(l)}, \boldsymbol{H}_{\mathcal{E}_T}) \qquad \boldsymbol{M}_{\mathcal{U}}^{(l)} = \Phi_{msg}^{\mathcal{V}\xrightarrow{\mathcal{E}_T}\mathcal{U}}(\mathcal{V}, \mathcal{E}_T, \boldsymbol{H}_{\mathcal{V}}^{(l)}, \boldsymbol{H}_{\mathcal{E}_T}) \qquad (1)$$

where $l$ is the number of current layer and $\Phi_{msg}^{\mathcal{U}\xrightarrow{\mathcal{E}_T}\mathcal{V}}$ is the message function which collects topological messages from *nets* $\mathcal{U}$ and sends them to *cells* $\mathcal{V}$ via *topo-edges* $\mathcal{E}_T$ ($\Phi_{msg}^{\mathcal{V}\xrightarrow{\mathcal{E}_T}\mathcal{U}}$ similarly). In Geometrical Message-passing, we consider the *geom-edges* $\mathcal{E}_G$ and collect geometrical messages for *cells* $\mathcal{V}$:

$$\boldsymbol{M}_{\mathcal{V}}^{(l),geom} = \Phi_{msg}^{\mathcal{V}\xrightarrow{\mathcal{E}_G}\mathcal{V}}(\mathcal{V}, \mathcal{E}_G, \boldsymbol{H}_{\mathcal{V}}^{(l)}, \boldsymbol{H}_{\mathcal{E}_G}) \qquad (2)$$

Then, we fuse the topological and geometrical messages and update the representations:

$$\boldsymbol{M}_{\mathcal{V}}^{(l)} = \text{MaxPooling}(\boldsymbol{M}_{\mathcal{V}}^{(l),geom}, \boldsymbol{M}_{\mathcal{V}}^{(l),topo}) \qquad (3)$$

$$\boldsymbol{H}_{\mathcal{V}}^{(l+1)} = \Phi_{update}(\boldsymbol{H}_{\mathcal{V}}^{(l)}, \boldsymbol{M}_{\mathcal{V}}^{(l)}) \qquad \boldsymbol{H}_{\mathcal{U}}^{(l+1)} = \Phi_{update}(\boldsymbol{H}_{\mathcal{U}}^{(l)}, \boldsymbol{M}_{\mathcal{U}}^{(l)}) \qquad (4)$$

where the update function $\Phi_{update}(\boldsymbol{H}, \boldsymbol{M}) = \boldsymbol{H} + \text{Tanh}(\boldsymbol{M})$.

The efficiency and the capability of perceiving heterogeneous information (e.g. the edge embeddings in $\mathcal{E}_T$ and $\mathcal{E}_G$ which encode topological and geometrical information) should be considered when designing the exact message functions $\Phi_{msg}$. In Topological Message-passing, inspired by [17], for a *net* $u$, we fuse the representations of surrounding *cells* $\{v|(v, u) \in \mathcal{E}_T\}$ and *topo-edges* connecting them:

$$\Phi_{msg}^{\mathcal{V}\xrightarrow{\mathcal{E}_T}\mathcal{U}}(\{(\boldsymbol{h}_v^{\mathcal{V}}, \boldsymbol{h}_{(v,u)}^{\mathcal{E}_T})|(v,u) \in \mathcal{E}_T\}) = \sum_{(v,u)\in\mathcal{E}_T} (\boldsymbol{W}_{\mathcal{E}_T\to\mathcal{U}}\boldsymbol{h}_{(v,u)}^{\mathcal{E}_T}) \odot (\boldsymbol{W}_{\mathcal{V}\to\mathcal{U}}\boldsymbol{h}_v^{\mathcal{V}}) \qquad (5)$$

where $\boldsymbol{h}_v^{\mathcal{V}}$ is the representation vector of *cell* $v$, $\boldsymbol{h}_{(v,u)}^{\mathcal{E}_T}$ is the representation vector of *topo-edge* connecting $v, u$, $\boldsymbol{W}_{\mathcal{V}\to\mathcal{U}}, \boldsymbol{W}_{\mathcal{E}_T\to\mathcal{U}}$ are learnable weight matrices and $\odot$ is the element-wise multiplication. As *topo-edge*'s representations have already been collected in $\Phi_{msg}^{\mathcal{V}\xrightarrow{\mathcal{E}_T}\mathcal{U}}$, we only consider *net*'s representations $\boldsymbol{h}_u^{\mathcal{U}}$ in the messages passed back to the *cells*, which speeds up message-passing (because of less computation) with minor topological information loss:

$$\Phi_{msg}^{\mathcal{U}\xrightarrow{\mathcal{E}_T}\mathcal{V}}(\{\boldsymbol{h}_u^{\mathcal{U}}|(v,u) \in \mathcal{E}_T\}) = \sum_{(v,u)\in\mathcal{E}_T} \boldsymbol{W}_{\mathcal{U}\to\mathcal{V}}\boldsymbol{h}_u^{\mathcal{U}} \qquad (6)$$

In Geometrical Message-passing, to enhance the geometrical information, the *geom-edge*'s representations are used to compute the edge weights when convolving the *cells*:

$$\Phi_{msg}^{\mathcal{V}\xrightarrow{\mathcal{E}_G}\mathcal{V}}(\{(\boldsymbol{h}_{v^*}^{\mathcal{V}}, \boldsymbol{h}_{(v,v^*)}^{\mathcal{E}_G})|(v,v^*) \in \mathcal{E}_G\}) = \sum_{(v,v^*)\in\mathcal{E}_G} (\boldsymbol{a}^\top\boldsymbol{h}_{(v,v^*)}^{\mathcal{E}_G}) \cdot \boldsymbol{W}_{\mathcal{V}\to\mathcal{V}}\boldsymbol{h}_{v^*}^{\mathcal{V}} \qquad (7)$$

where $\boldsymbol{a}$ is a learnable weight vector and $\boldsymbol{W}_{\mathcal{V}\to\mathcal{V}}$ is a learnable weight matrix.

**Discussion of Inference Time.** Assume that the hidden layer dimensions of *cell*, *net*, *topo-edge*, *geom-edge* are $F_{\mathcal{V}}, F_{\mathcal{U}}, F_{\mathcal{E}_T}, F_{\mathcal{E}_G}$. The inference times of $\Phi_{msg}^{\mathcal{V}\xrightarrow{\mathcal{E}_T}\mathcal{U}}, \Phi_{msg}^{\mathcal{U}\xrightarrow{\mathcal{E}_T}\mathcal{V}}, \Phi_{msg}^{\mathcal{V}\xrightarrow{\mathcal{E}_G}\mathcal{V}}$ are $O(|\mathcal{E}_T|(F_{\mathcal{E}_T}F_{\mathcal{U}} + F_{\mathcal{V}}F_{\mathcal{U}} + F_{\mathcal{U}})), O(|\mathcal{E}_T|F_{\mathcal{U}}F_{\mathcal{V}}), O(|\mathcal{E}_G|(F_{\mathcal{E}_G} + F_{\mathcal{V}}^2))$, respectively. The time complexities of fusing (MaxPooling) and updating are $O(|\mathcal{V}|F_{\mathcal{V}}), O(|\mathcal{V}|F_{\mathcal{V}} + |\mathcal{U}|F_{\mathcal{U}})$, respectively. As

the dimensions are constant numbers and $|\mathcal{E}_T| = O(|\mathcal{P}|), |\mathcal{E}_G| = O(|\mathcal{V}|)$ (see Appendix B.2), the total inference time in one Topo-Geom Message-passing layer is $O(|\mathcal{V}| + |\mathcal{U}| + |\mathcal{P}|)$, which is linear to the scale of input Circuit Graph. Therefore, our message-passing method can handle Circuit Graph **with optimal efficiency**, and it is further demonstrated to be as fast as **CongestionNet**[5], a deep GAT architecture, in Tab.1.

### 4.3 Task-adaptive Readout

After $L$ iterations of message-passing, we read out the *cell* and *net* representations $\boldsymbol{H}_{\mathcal{V}}^{(L)}, \boldsymbol{H}_{\mathcal{U}}^{(L)}$ to handle diverse downstream tasks on circuits. For tasks on Cell-level (e.g. congestion prediction for each *cell* [5]), to enhance the raw features, we concatenate the *cell* representation and its raw features and pass them through an MLP (similarly for Net-level tasks e.g. net length prediction [12]):

$$\hat{\boldsymbol{y}}_{\text{cell}} = \text{MLP}(\boldsymbol{H}_{\mathcal{V}}^{(L)} \oplus \boldsymbol{X}_{\mathcal{V}}) \qquad \hat{\boldsymbol{y}}_{\text{net}} = \text{MLP}(\boldsymbol{H}_{\mathcal{U}}^{(L)} \oplus \boldsymbol{X}_{\mathcal{U}}) \tag{8}$$

However, Grid-level tasks (e.g. congestion prediction for each *grid* in chip map [9]) assign targets on each *grid*, which our model is not aware of. To enable model's training and evaluation on these tasks, we generate an output representation on each *grid* by mean-pooling the representation of *cells* inside the *grid*:

$$\hat{\boldsymbol{y}}_{\text{grid}} = \text{MLP}(\hat{\boldsymbol{M}} \boldsymbol{H}_{\mathcal{V}}^{(L)}) \tag{9}$$

where $\hat{\boldsymbol{M}} \in \mathbb{R}^{C_x \times C_y \times |\mathcal{V}|}$ is the transformation matrix with $\sum_k \hat{\boldsymbol{M}}_{i,j,k} = 1, \forall i, j$.

## 5 Experiments

### 5.1 Tasks and Datasets

**Congestion Prediction** is the task of predicting the routing congestion before the wires are routed in the detailed routing stage. It is widely used in placement tools to provide quick feedback about the quality of placement and avoid placement solutions with poor routability [18][19][20]. In order to identify and solve potential congested structures earlier, multiple works have attempted to predict cell-level congestion in logic synthesis stage, before cells are placed [6][21][22]. We conduct the experiment on ISPD2011[3], which contains 12 VLSI designs in total. We use 10 designs (1/2/3/5/6/7/9/11/14/16) for training, design #18 for validation, and #19 for testing. We use DREAMPlace[18] to place cells and initialize the raw features of *cells*, *nets* and *grids*. NCTU-GR 2.0[23], a popular global router, is used to generate the congestion targets on the grids. The congestion target of each *cell* is set as the value of the grid it is located in. For congestion prediction in logic synthesis stage, we only use the topology of the circuits and the geometry-insensitive features. For prediction in placement, we additionally use the *cells*' positions generated by DREAMPlace.

Similar to [6], we compare the prediction and ground-truth in Pearson/Spearman/Kendall correlation on both Cell-level and Grid-level. We also divide the congestion values to $[0, 0.9]$ and $(0.9, \infty)$ and use precision/recall/F1-score to further evaluate their ability of identifying congestion [14].

**Net Wirelength Prediction** aims to deduce the wire length of each *net*, which is an important indicator of the eventual chip performance [24]. We use half-perimeter wirelength (HPWL) as the wirelength estimator, which is the most commonly used method for wirelength calculation [2]. We conduct the experiment on DAC2012[4], where we use 7 designs (3/6/7/9/11/12/14) for training, design #16 for validation and #19 for testing. DREAMPlace[18] is used to generate the targets for every *net*. The featurization of circuits in both stages is the same with **Congestion Prediction** mentioned above.

As the regression targets of wirelength range from 0 to about 25k, we take the $\log_{10}$ of them to make the distribution of targets more smoothing. We evaluate the results in Pearson/Spearman/Kendall correlation as well as Mean Average Error (MAE) and Root Mean Square Error (RMSE).

**Transfer Task** is designed to further evaluate the representativeness of extracted GNN features. Here, we first train Circuit GNN/LHNN with Congestion Prediction task and then evaluate/fine-tune them with Wirelength Prediction.

---

[3]http://www.ispd.cc/contests/11/ispd2011_contest.html
[4]http://archive.sigda.org/dac2012/contest/dac2012_contest.html

## 5.2 Baselines and Settings

When conducting the two tasks above, we compare our model with the following baselines. Traditional machine learning methods include **MLP**, typical graph representation models **GCN**[25]/**GAT**[11]/**GraphSAGE**[26]/**MPNN**[10] and **pix2pix**[9]/, a typical image translation model in CV. We also carry out results on EDA-customized machine learning models: (1) **CongestionNet**[5], a multi-layer graph attentive architecture designed to predict circuit congestion; (2) **Net²ᶠ/Net²ᵃ**[12], pre-placement net representation models with customized GNN; (3) **LHNN**[14]: a geometrical congestion prediction method supplied with topological information i.e. net span. To further validate our model's ability to fuse the topological and geometrical information, we test two variants of our model: **Ours (w/o. geom.)** which throws all *geom-edges* and **Ours (w/o. topo.)** which throws all *topo-edges* in Circuit Graph (Def.3).

For cited methods, we use their default model settings. For Circuit GNN, we set hidden layer dimensions of *cell*, *net*, *topo-edge*, *geom-edge* $(F_\mathcal{V}, F_\mathcal{U}, F_{\mathcal{E}_T}, F_{\mathcal{E}_G}) = (64, 128, 8, 4)$ and message passing layers $L = 2$. We have default settings of window size $(w_x, w_y) = (32, 40)$ and link capacity $c = 5$, and their parameter sensitivity is tested in Appendix C to show the robustness of our model. When training our model with `Adam Optimizer`, we use learning rate $\gamma = 0.0002$, learning rate decay $\Delta\gamma = 0.02$, weight decay $\eta = 0.0002$ and training epoch $e = 100$.

## 5.3 Result of Congestion Prediction

Table 1: Congestion prediction result in logic synthesis stage

| Baseline | Time (s/epoch) | Cell-level | | | Grid-level | | |
|---|---|---|---|---|---|---|---|
| | | pearson | spearman | kendall | pearson | spearman | kendall |
| GCN | 9.43 | 0.777 | 0.265 | 0.199 | 0.221 | 0.366 | 0.260 |
| GraphSAGE | 11.79 | 0.776 | 0.252 | 0.188 | 0.208 | 0.375 | 0.268 |
| GAT | 13.90 | 0.777 | 0.267 | 0.200 | 0.215 | 0.399 | 0.280 |
| CongestionNet | 22.31 | 0.777 | 0.269 | 0.200 | 0.277 | 0.394 | 0.280 |
| MPNN | 116.24 | **0.780** | **0.289** | **0.217** | 0.292 | 0.458 | 0.319 |
| Ours (w/o. geom.) | 21.62 | 0.779 | **0.289** | **0.217** | **0.315** | **0.468** | **0.329** |

Table 2: Congestion prediction result in placement stage (in correlation)

| Baseline | Time (s/epoch) | Cell-level | | | Grid-level | | |
|---|---|---|---|---|---|---|---|
| | | pearson | spearman | kendall | pearson | spearman | kendall |
| GAT (w. geom.) | 16.21 | 0.777 | 0.263 | 0.197 | 0.210 | 0.397 | 0.279 |
| pix2pix | 4.46 | - | - | - | 0.562 | 0.554 | 0.392 |
| LHNN | 305.47 | - | - | - | **0.703** | 0.695 | 0.540 |
| Ours (w/o. topo.) | 21.54 | 0.883 | 0.713 | 0.573 | 0.684 | 0.730 | 0.536 |
| Ours | 27.07 | **0.887** | **0.714** | **0.575** | 0.697 | **0.770** | **0.577** |

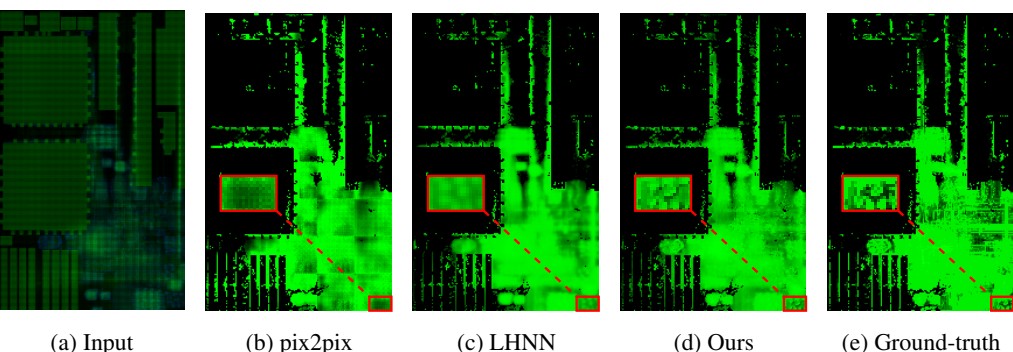

(a) Input      (b) pix2pix      (c) LHNN      (d) Ours      (e) Ground-truth

Figure 5: Visualization of congestion maps of circuit `ispd2011/superblue19` produced by the baselines.

We first evaluate the congestion prediction result on circuits in logic synthesis stage when geometric information is not available (Tab.1). Then we perform the same task in placement stage as shown in Tab.2 and Appendix Tab.12. Note that geometrical methods **pix2pix** and **LHNN** are not aware of *cells* in circuit design, so they are not evaluated on Cell-level. **GAT (w. geom.)** is regular GAT with *cell* positions as additional features.

The results show that: (1) In logic synthesis stage, our method with only *topo-edges* achieves the best performance (16.7% over cutting-edge **CongestionNet**) with a similar time cost compared to traditional GNN models (5x faster than **MPNN** which has a time-expensive and underused edge function for netlist input). (2) In placement stage, our method beats the cutting-edge **LHNN** in most metrics (5.6% on average) while taking only one-tenth of the run-time. The superior performance of our model is primarily attributed to the fusion of both topological and geometrical information.

Fig.5 visualizes the predicted congestion values using different methods. Compared to vision-based method [9] and lattice network-based method [15], our proposed method can generate finer congestion prediction with better discriminability.

## 5.4 Result of Net Wirelength Prediction

Table 3: Net wirelength prediction in placement stage (↓ means "lower is better")

| Baseline | Time (s/epcoh) | pearson | spearman | kendall | MAE↓ | RMSE↓ |
|---|---|---|---|---|---|---|
| MLP | 2.22 | 0.493 | 0.547 | 0.415 | 0.626 | 0.819 |
| Net$^{2f}$ | 10.42 | 0.517 | 0.635 | 0.525 | 0.615 | 0.825 |
| Net$^{2a}$ | 19.83 | 0.632 | 0.656 | 0.553 | 0.614 | 0.821 |
| LHNN | 260.00 | 0.801 | 0.796 | 0.603 | 0.581 | 0.780 |
| Ours | 14.79 | **0.848** | **0.835** | **0.646** | **0.483** | **0.683** |

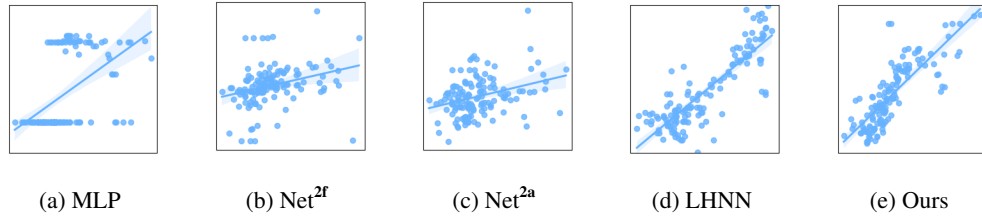

(a) MLP  (b) Net$^{2f}$  (c) Net$^{2a}$  (d) LHNN  (e) Ours

Figure 6: Scattering the models' output (axis-y) and ground-truth (axis-x) (placement stage).

The result of net wirelength prediction is shown in Appendix Tab.11 (logic synthesis stage), Tab.3 and Fig.6 (placement stage). We obtain LHNN's result on this task by reading out its net representations generated in the intermediate stage. The result shows that our model achieves SOTA performance (16.9% error gain) with similar time cost to **Net$^{2f}$** and **Net$^{2a}$**.

## 5.5 Result of Transfer Task

Here are the experimental settings:

- For the **evaluation** setting, a different readout module is trained from scratch, but GNN's parameters are fixed.

- For the **fine-tuning** setting, a different readout module is trained from scratch, and GNN's parameters are fine-tuned at the same time.

- For **evaluation & fine-tuning**, LHNN and Ours are only trained for 1/5 epochs of default setting, to show the transferability of learned features.

Table 4: Transfer experiment from congestion prediction to wirelength prediction. Results are evaluated in Grid-level.

| Baseline | Time (s/epoch) | pearson | spearman | kendall |
|---|---|---|---|---|
| MLP | 2.22 | 0.493 | 0.547 | 0.415 |
| LHNN (evaluate) | 192.45 | 0.689 | 0.715 | 0.563 |
| Ours (evaluate) | 9.55 | **0.799** | **0.811** | **0.622** |
| LHNN (fine-tune) | 248.96 | 0.805 | 0.794 | 0.612 |
| Ours (fine-tune) | 14.8 | **0.842** | **0.829** | **0.639** |
| LHNN | 260 | 0.801 | 0.796 | 0.603 |
| Ours | 14.79 | 0.848 | 0.835 | 0.646 |

The results show that the knowledge Circuit GNN learns from Congestion Prediction can be easily transferred to another task (no matter for direct use or fine-tuning), while LHNN is weak in transferability.

## 6 Conclusion and Future Work

We present a versatile graph neural network to facilitate EDA circuit design process. To this end, we design a heterogeneous graph, Circuit Graph, to integrate topological and geometrical information into a unified data structure, based on which we further propose a message-passing and fusion approach named Circuit GNN. It is the first circuit representation method applied to multiple EDA tasks and stages. By integrating multi-source information, Circuit Graph outperforms previous methods in prediction performance and computation time. Our work further supports the EDA process "shift-left", a new future direction that aims to speed-up circuit design by deeply combining artificial intelligence in all EDA tool chains.

**Limitations** Although AI for EDA becomes a hot research topic recently and some deep learning-driven techniques have been adopted in the main-stream EDA tools (Cadence, Synopsys, etc.), there is still a gap between the novel machine learning algorithms and their application in commercial tools. Moreover, in early EDA stages, other circuit representations like data-flow graph or And-Inverter Graph (AIG) graph might be used. As the meaning of nodes and edges in these graphs are different from netlist graph used in the paper, our method might not apply to these graphs in early EDA stages.

**Societal Impact.** Our solution explores dual-stage circuit representation to serve various EDA downstream tasks, which have limited societal impact. Whatever, there is a minimal possibility of misuse that violate some ethics of life science.

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
