# OpenReview forum: "Versatile Multi-stage Graph Neural Network for Circuit Representation"
_NeurIPS.cc/2022/Conference — NeurIPS 2022 Accept_

### Official Review · Reviewer_73KN · 2022-07-11

**Rating:** 6
**Confidence:** 4
**Soundness:** 2 fair
**Presentation:** 3 good
**Contribution:** 2 fair

**Summary:**

The authors propose a unified way to construct graphs in different phases of EDA flow, and develop a general GNN model for downstream EDA tasks. Specifically, the proposed approach first constructs a heterogeneous graph by incorporating cell-cell connections (geometrical information) and cell-net connections (topological information). The node and edge features are generated based on physical properties of cells, pins, and nets. Then, a circuit GNN model is proposed to apply message passing on cell-cell and cell-net connections separately, which produces the representations of cells and nets for downstream tasks. The experimental results show that the proposed method increases 16.7% accuracy on congestion prediction and reduces 16.9% error on wirelength prediction.

**Questions:**

  - For the wirelength prediction task in placement stage, authors use DREAMPlace to generate both the ground truth and node/edge features. If the ground truth is already known, why does it even need to generate features for GNN?
  - Is there any reason/insight why two different message-passing functions are used for topo-edges (Equation 5) and geom-edges (Equation 7)?
  - Apart from logic synthesis and placement, how does the proposed work incorporate graphs in other EDA stages (e.g., data-flow graph in high-level synthesis)?
  - Is it necessary to have geom-edges? Can we encode the cell positions into node feature vectors instead?

**Limitations:**

Thanks authors for mentioning potential limitations of this work. One key challenge of deploying ML models into commercial EDA tools is the model generalizability. Authors can evaluate the trained model on more unseen designs to see if it is truly generalizable.

**Strengths And Weaknesses:**

**Key Strength**

  - The paper is clearly written. All the technical steps are easy to follow.
  - The proposed method can be used to solve multi-stage EDA tasks.


**Key Weakness**

Although the proposed circuit graph construction and GNN model are all reasonable, they lack some technical significance. For example,
  - For circuit graph construction, it is straightforward to construct a bipartite graph based on cell-net connections from netlist, in order to produce representations of cells and nets for downstream tasks. Hence, the contribution is limited for the graph construction, especially in logic synthesis stage where placement information is not available.
  - For GNN model, it is a common way to apply message passing individually per edge type for handling heterogeneous graphs (e.g., [2]). Thus, the novelty of the proposed model is limited.


Although the experiments show promising accuracy gains for downstream EDA tasks, further clarification could make the improvements more convincing:
  - Missing strong GNN baselines: The chosen baselines (i.e., GCN, GraphSAGE, and GAT) only consider node features. Since edge features are important in this paper, authors should compare the proposed model against stronger baselines (e.g., MPNN[1]) that incorporate edge features, on the same input graph. Without a stronger baseline, the contribution of the proposed GNN model is unclear.
  - Not tuning hyperparameters for baselines: Authors choose the default hyperparameters for baselines from their original papers. Since the datasets used in those papers (e.g., [3]) are different from this paper, hyperparameter tuning is necessary.
  - Not comparing against DREAMPlace: The purpose of wirelength prediction is to speedup EDA design closure. Nonetheless, there are no results of the runtime comparison between the proposed model and the placement method DREAMPlace, which is a very fast placement method by exploiting GPUs. Without this comparison, it's unclear about the motivation of wirelength prediction in placement.

[1]: Gilmer et al. "Neural message passing for quantum chemistry." ICML'17. \
[2]: Zhang et al. "Heterogeneous graph neural network." KDD'19. \
[3]: Xie et al. "Pre-Placement Net Length and Timing Estimation by Customized Graph Neural Network." TCAD'22.

---

> ### Author Response · Authors · 2022-07-31
> **Official Reply to Reviewer 73KN PART 3/3**
>
> ## About Your Questions
>
> 1. DREAMPlace is a publicly available and easily setup toolkit for researchers studying VLSI placement problem. In our paper, for all the tasks (congestion prediction and wirelength prediction), we use DREAMPlace to help generate features and ground truth labels to construct learning based EDA tasks, where deep learning methods are expected to learn the good mapping function from the input features to the ground truth labels using the generated data. This setting is following the experiment settings in [1].
> 2. On one hand, there are usually more geom-edges than topo-edges in Circuit Graph (3.4M geom-edges and 1.9M topo-edges in *superblue19*[2]), so for geom-edges we prefer edge-weight summation rather than inner product, which is $F_{\cal U}$ (hidden dimension of net) times more expansive in computation.  On the other hand, it is also reasonable for geom-edges to use edge-weight summation because geometrically closer cells have a stronger relationship. Still, we test the performance when topo-edges use edge-weight summation or geom-edges use the inner product:
>
>
>     |  | time | pearson (N) | spearman (N) | kendall (N) | pearson (G) | spearman (G) | kendall (G) |
>     | --- | --- | --- | --- | --- | --- | --- | --- |
>     | topo w. edge-weight summation | 25.35 | 0.886 | 0.707 | 0.570 | 0.694 | 0.743 | 0.552 |
>     | geom w. inner product | 37.71 | 0.886 | **0.717** | **0.579** | 0.689 | 0.734 | 0.542 |
>     | Ours | 27.07 | **0.887** | 0.714 | 0.575 | **0.697** | **0.770** | **0.577** |
>
>     We can observe that:
>
>     - If we use edge-weight summation to process topo-edges, time is marginally saved but performance gets worse.
>     - If we use the inner product to process geom-edges, time cost will increase with no significant improvement.
> 3. Apart from logic synthesis and placement, the proposed work can be potentially used to predict timing in routing stage or Clock Tree Synthesis (CTS) outcomes like clock power, max skews, etc. Some prior works have attempted to use machine learning-based methods for these tasks [4][5][6]. However, in data flow graphs in high level synthesis, the nodes represent computation functions and the directed edges represent data path, which are different from the meaning of nodes and edges in a circuit netlist. The method proposed in our work might not apply to data flow graph.
> 4. Directly encoding the cell positions as features leads to very bad generalization because raw 3D positions do not satisfy translation and rotation invariances [3]. Here are some experiments: (Ours (w/o. geom.) (pos. encode) is the modification we made which encodes the cell positions into node features instead.)
>
>
>     |  | time | pearson (N) | spearman (N) | kendall (N) | pearson (G) | spearman (G) | kendall (G) |
>     | --- | --- | --- | --- | --- | --- | --- | --- |
>     | GAT | 13.90 | 0.777 | 0.267 | 0.200 | 0.215 | 0.399 | 0.280 |
>     | GAT (pos. encode) | 16.21 | 0.777 | 0.263 | 0.197 | 0.210 | 0.397 | 0.279 |
>     | Ours (w/o. geom.) | 21.62 | 0.779 | 0.289 | 0.217 | 0.315 | 0.468 | 0.329 |
>     | Ours (w/o. geom.) (pos. encode) | 22.55 | 0.766 | 0.328 | 0.292 | 0.228 | 0.475 | 0.411 |
>     | Ours | 27.07 | **0.887** | **0.714** | **0.575** | **0.697** | **0.770** | **0.577** |
>
>     We can see that directly encoding the cell positions cannot improve overall performance, while geom-edges do.
>
>
> ## Acknowledgment
>
> Thank you again for your constructive suggestions, which help us find some points we didn’t explain clearly. We will get them clarified in our later version.
>
> [1] Ghose et al. Generalizable Cross-Graph Embedding for GNN-based Congestion Prediction. In ICCAD 2021
>
> [2] Bustany et al. ISPD 2015 Benchmarks with Fence Regions and Routing Blockages for Detailed-Routing-Driven Placement.
>
> [3] Yang et al. Deep Molecular Representation Learning via Fusing Physical and Chemical Information. In NeurIPS 2021
>
> [4] Barboza et al. Machine Learning-Based Pre-Routing Timing Prediction with Reduced Pessimism. In DAC 2019.
>
> [5] Yang et al. Pre-Routing Path Delay Estimation Based on Transformer and Residual Framework. In ASP-DAC 2022.
>
> [6] Lu et al. GAN-CTS: A Generative Adversarial Framework for Clock Tree Prediction and Optimization. In ICCAD 2019.
>
> [7] Xie et al. Net^2^: A Graph Attention Network Method Customized for Pre-Placement Net Length Estimation. In ASPDAC 2021
>
> [8] Dai et al. NCTU-GR: Efficient Simulated Evolution-Based Rerouting and Congestion-Relaxed Layer Assignment on 3-D Global Routing. In TVLSI 2010
>
> [9] Zhang et al. Heterogeneous graph neural network. In KDD 2019

---

> > ### Comment · Reviewer_73KN · 2022-08-08
> > **Follow-up on authors' response**
> >
> > Thank you for your detailed responses. My remaining concerns are listed below:
> >
> > **Follow-up on the first question:**
> >
> > Since DreamPlace is used to produce ground truth, why do we even need to generate node features by DreamPlace and then apply GNN? In other words, we already know the ground truth once we obtain node features by DreamPlace, so what's the point of using GNN to predict the ground truth again?
> >
> > **Follow-up on the third question:**
> >
> > If the proposed approach cannot work on data-flow graphs in the high-level synthesis stage, authors should clearly mention this limitation in the paper, instead of claiming that the proposed approach is compatible across EDA stages.

---

> > > ### Author Response · Authors · 2022-08-09
> > > **Authors' Response**
> > >
> > > ## For generating both features and labels with DREAMPlace
> > >
> > > DREAMPlace is an open platform which collects lots of EDA tools e.g. RUDY, NCTUgr. Many AI4EDA works[1] [2] [3] use these tools to generate raw features and labels to construct machine learning tasks, where the DL models are evaluated. It's also common in other machine learning fields besides of AI4EDA. Take molecular machine learning as example. The researchers usually use RDKit(https://rdkit.org/), a open-source cheminformatics software to generate **atom/bond raw features as well as targets (e.g. conformation and energy)** [4] [5] [6], while none of them argue that the experiments are meaningless if these data come from the same toolkit.
> > > Therefore, it is universally acknowledged to conduct the experiments with features and labels from non-AI tools and **check how efficiently and effectively DL models can learn from raw features and handle the tasks**.
> > >
> > > ## For other stages like data-flow
> > > We claim in the paper that our method is compatible across multiple EDA tasks and stages. This claim is backed by the experiment results of two EDA tasks (congestion prediction and net wirelength prediction) at two EDA stages (logic synthesis and placement). The input of our model is circuit netlist, which is a collection of electronic components connected by physical wires. Netlist is only available after technology mapping in logic synthesis. However, there are other circuit representations in early EDA stages like data-flow graph in high-level synthesis or And-Inverter Graph (AIG) in early logic synthesis stage. Our method might not apply to these graphs as their nodes and edges have different meaning. To make the claim more accurate, we update the limitation section of the paper with following:
> > > "In early EDA stages, other circuit representations like dataflow graph or And-Inverter Graph (AIG) graph might be used. As the meaning of nodes and edges in these graphs are different from netlist graph used in the paper, our method might not apply to these graphs used in early EDA stages."
> > >
> > > [1] Ghose et al. Generalizable Cross-Graph Embedding for GNN-based Congestion Prediction. In ICCAD 2021
> > >
> > > [2] Wang et al. LHNN: Lattice Hypergraph Neural Network for VLSI Congestion Prediction. In DAC 2022
> > >
> > > [3] Xie et al. Net2: A Graph Attention Network Method Customized for Pre-Placement Net Length Estimation. In ASPDAC 2021
> > >
> > > [4] Xiong et al. Pushing the Boundaries of Molecular Representation for Drug Discovery with the Graph Attention Mechanism. In Med Chem 2020 Aug 27;63(16)
> > >
> > > [5] Li et al. Conformation-Guided Molecular Representation with Hamiltonian Neural Networks. In ICLR 2021
> > >
> > > [6] Yang et al. Deep Molecular Representation Learning via Fusing Physical and Chemical Information. In NeurIPS 2021

---

> > > > ### Comment · Reviewer_73KN · 2022-08-09
> > > > **Follow-up on authors' response**
> > > >
> > > > Thanks for your response and additional results.
> > > >
> > > > Regarding to the support of EDA stages, this is the sentence copied from the Introduction section "To our best knowledge, this is the first unified circuit representation approach that can be easily compatible across EDA tasks and stages.", which gives people an impression that the proposed approach works for all EDA stages. I would encourage authors to rephrase this sentence in the next revision to avoid misleading.
> > > >
> > > > Overall, most of my major concerns are addressed after rebuttal, though a few minor concerns still exist (e.g., runtime speedup is too marginal for the wirelength prediction task in placement stage). Considering the contribution of this work to both GNN and EDA communities, I raise my score to 6. Authors are strongly encouraged to include their rebuttal responses (e.g., all the additional experiments such as MPNN and clarifications such as compatibility across EDA stages) in the final version.

---

> ### Author Response · Authors · 2022-07-31
> **Official Reply to Reviewer 73KN PART 2/3**
>
> ## Experiments
>
> 1. **MPNN.** Beyond traditional GNNs (GCN et al.), we also compared our model against various advanced GNNs specially designed for EDA problems, e.g. CongestionNet, Net^2^ and LHNN. We considered comparing it with MPNN, but MPNN is originally designed to handle small molecules rather than VLSI. The molecular graphs are small in scale but embed rich semantics in bonds (edges), so it is reasonable for MPNN to use a time-expensive edge message function ($O(H_v^2H_e)$, slower than our $O(H_vH_e)$). However, MPNN seems to be costly and underused when applied to large circuits with millions of nodes and edges. Here are some results: (Note that the hidden dimensions are the same across baselines. **N** refers to Node-level and **G** refers to Grid-level)
>
>
>     |  | time | pearson (N) | spearman (N) | kendall (N) | pearson (G) | spearman (G) | kendall (G) |
>     | --- | --- | --- | --- | --- | --- | --- | --- |
>     | GCN | 9.43 | 0.777 | 0.265 | 0.199 | 0.221 | 0.366 | 0.260 |
>     | Ours (w/o. geom.) (MPNN conv.) | 116.24 | **0.780** | **0.289** | **0.217** | 0.292 | 0.458 | 0.319 |
>     | Ours (w/o. geom.) | 21.62 | 0.779 | **0.289** | **0.217** | **0.315** | **0.468** | **0.329** |
>
> 1. **Hyperparameters.** We tried to tune the hyperparameters in Net^2^ and LHNN (e.g. the hidden dimension, layer number, and edge sample number), but there has no overall improvement, so we kept the default settings. For example, the influence of the hidden dimension of LHNN shows below:
>
>
>     | hidden dim. | time | pearson (G) | spearman (G) | kendall (G) |
>     | --- | --- | --- | --- | --- |
>     | 16 | 205.05 | 0.703 | 0.698 | **0.545** |
>     | 32 (default) | 305.47 | 0.700 | **0.701** | 0.540 |
>     | 64 | 340.42 | **0.707** | 0.696 | 0.541 |
> 2. **DREAMPlace Comparison.** The inference time of our model is 178x faster than DREAMPlace (1.6s vs. 285s) when predicting net length for circuit **superblue19** in logic synthesis stage (see Table 10), as DREAMPlace should do placement first before adopting HPWL to calculate net length.
>
>     Besides, we use DREAMPlace, a popular non-AI toolkit, to generate the labels with HPWL to construct Net Length Prediction task, similar to the experiment settings in [7]. The key point of conducting this experiment is to show that the representation learned by our model can be efficiently and effectively applied to multiple circuit stages and EDA tasks. In contrast, the existing DL models are not as versatile.

---

> > ### Comment · Reviewer_73KN · 2022-08-08
> > **Follow-up on authors' response**
> >
> > Thank you for your detailed responses. My remaining concerns are listed below:
> >
> > **MPNN**
> >
> > I would encourage authors to include MPNN results in the next revision and highlight the advantage of the proposed model over MPNN, which would make the paper stronger.
> >
> > **DreamPlace comparison**
> >
> > I could not find the results that authors mentioned (1.6s vs. 285s) in Table 10. More importantly, I was asking the runtime comparison between DreamPlace and the proposed approach **in the placement stage**, rather than the logic synthesis stage. If there are no such results to show the runtime speedup, it is unclear about the motivation why we need to apply ML-based methods for net wirelength prediction in the placement stage.

---

> > > ### Author Response · Authors · 2022-08-09
> > > **Authors' Response**
> > >
> > > ## For MPNN
> > >
> > > Thank you for your suggestion. We will include our discussion about the advantages of our proposed method over other GNN baselines that incorporate edge features (MPNN) into an improved future version.
> > >
> > > ## For results in Table 10
> > >
> > > We are sorry for this mistake. The original version of Table 10 only includes the training time of the baselines. We have submitted an updated version and included the inference time comparison results we mentioned between ours and DREAMPlace on *superblue19* (1.6s vs. 285s for the logic synthesis phase, and **3.51s vs. 3.63 in placement stage**) in Appendix F.
> > >
> > > ## For the runtime of DREAMPlace in net length prediction
> > >
> > > Since the reviewer explicitly ask for the runtime comparison of DREAMPlace [1], we believe it is necessary to fully explain the working mechanism of DREAMPlace in the global placement stage and its relation to our proposed Circuit GNN predictor. DREAMPlace is a convenient open platform placer to solve region-constrained global placement problems. It proposes a neural network-based analytical solution to solve the global placement problems efficiently. To provide the feedback signal of the quality of the current design layout, there are several metrics that are important to consider, including wirelength, congestion and etc. DREAMPlace has integrated famous global routers such as NCTUgr [2] and Rudy [3] to provide an estimation of their design layout after each training step in their analytical solution.
> > >
> > > However, the existing accurate estimation of the congestion given the current layout is expensive, such as NCTUgr [2], which makes the feedback signal generation process to become the efficiency bottleneck of the global placement stage. This aspect provides the motivation for the following learning-based algorithm[3] [4] [5] to learn a proxy function and provides a relatively accurate estimation of routing congestion values with a much faster runtime. One work follows the same motivation.
> > >
> > > In our paper, we did not claim our method is able to replace the DREAMPlace framework since our proposed method is only able to predict the metrics of interest such as congestion and wirelength. It is unable to replace the placer tool such as DREAMPlace. For the global placement stage, the runtime improvement we claim over DREAMPlace is coming from using the proxy predictor over the classic congestion estimation tool when we are generating the quality feedback signal for each design layout. In the global placement stage, on the largest *superblue19* circuit, our inference time for generating the congestion label is around 4.09s and 3.51s for wirelength label (in total 7.6 s for each iteration step) while using the conventional NCTUgr and HPWL to estimate the ground truth label will take 55.2s and 3.63 s respectively (in total 58.8 s for each iteration step), which is 7.7 times slower than our proposed Circuit GNN per each iteration step.
> > >
> > > Indeed, on merely the wirelength prediction task, our runtime on *superblue19* is only marginally improved HPWL. However, the wirelength prediction is only a support task in the placement stage since the way we are representing the netlist and layout geometry information in our proposed Circuit Graph contains the **Nets level** representation. Thus, with a simple pooling layer and a decoder, we can easily obtain the wirelength prediction label. The wirelength predictor we proposed is not where the main source of runtime speedup comes from. Its effectiveness is more convincing in the logic synthesis stage as we do not even need to rely on the post placement layout information to provide a rough estimation in the earlier design stage.
> > >
> > > [1] Jiaqi Gu et al. DREAMPlace 3.0: Multi-Electrostatics Based Robust VLSI Placement with Region Constraints. In ICCAD 2020.
> > >
> > > [2] K. Dai, W. Liu, and Y. Li. NCTU-GR: Efficient simulated evolution-based rerouting and congestion-relaxed layer assignment on 3-d global routing. IEEE TVLSI, vol. 20, no. 3, pp. 459-472, 2012.
> > >
> > > [3] P. Spindler and F. M. Johannes. Fast and accurate routing demand estimation for efficient routability-driven placement. in Design, Automation & Test in Europe Conference & Exhibition, 2007.
> > >
> > > [4] R. Kirby, S. Godil, R. Roy, and B.
> > > Catanzaro. Congestionnet: Routing congestion prediction using deep graph neural networks. in VLSI 2019.
> > >
> > > [5] Ghose et al. Generalizable Cross-Graph Embedding for GNN-based Congestion Prediction. In ICCAD 2021.
> > >
> > > [6] Wang et al. LHNN: Lattice Hypergraph Neural Network for VLSI Congestion Prediction. In DAC 2022.

---

> ### Author Response · Authors · 2022-07-31
> **Official Reply to Reviewer 73KN PART 1/3**
>
> We greatly appreciate your careful and detailed review. Here are some points we would like to clarify:
>
> ## Justification for our Technical Significance
>
> The main technical significances of our solution are VERSATILITY and EFFICIENCY, which encourage us to find a **simple, general, compatible but effective** solution for netlist representation in the EDA field. To allow the proposed solution to have potential practical and commercial values, it will require the learning-based solution to achieve a significantly faster inference time compared to the traditional EDA routing tool, as well as a reasonable training time. For example, on the largest circuit we used in our paper (*superblue19*), estimating the congestion for each layout using the conventional tool NCTUGR [8] will take around 51 s. A learning-based method should have a significantly faster inference time (4s) to have a convincing argument to replace the conventional EDA tools. This particular requirement limits us from designing complicated and inefficient components in our model. For example, computationally more demanding meta path based Heterogeneous graph neural network method [9] suggested by the the reviewer might be potentially less favorable for this reason.
>
> The Circuit Graph and Circuit GNN are designed to guarantee VERSATILITY and EFFICIENCY, as we state below:
>
> ### Justification for our netlist graph construction (Circuit Graph)
>
> **FOR VERSATILITY.** We use topo-edges and geom-edges to capture the topology and geometry, which determine most of the properties of a circuit. For example, circuit congestion is mainly caused by crowdedly-placed cells (geometry) and dense nets (topology), while net length is determined by cells’ connectivity via nets (topology) and the distances among connected cells (geometry). Therefore, our Circuit Graphs can be used as inputs for diverse EDA prediction tasks.
>
> To **generalize** the geometrical information among circuits and tasks, we use cell-pair distances as the raw features of geom-edges, which **are invariant to translation and rotation** [3]. Besides, the distances can cover most of the geometrical information the models need to handle the tasks (e.g. the crowdedly-placed cells and distances among connected cells mentioned above).
>
> For circuits in logic synthesis stage, the Circuit Graph can **compatibly** process them by masking the geom-edges, where most of the topological information can still be preserved and used to handle various tasks.
>
> **FOR EFFICIENCY.** We accelerate the construction of geom-edges with shift-windows. Even VLSI with millions of cells and nets can be converted to Circuit Graph with a time cost linear to its scale (see Appendix B.2).
>
> ### Justification for our GNN model (Circuit GNN)
>
> **FOR VERSATILITY.** To capture the deep topological and geometrical relationships and generate informative representations for diverse EDA tasks, we not only pass the massages through topo-edges and geom-edges, but also **FUSE** them at the end of each layer.
>
> For input Circuit Graphs with no geom-edges (logic synthesis stage), the Circuit GNN can compatibly process them by masking the geometrical message-passing, where most of the topological information can still be preserved in output representations to serve downstream tasks.
>
> **FOR EFFICIENCY.** We use simple but effective message functions and fusion strategy to collect the topological and geometrical information. See more details in **Discussion of Inference Time** at the end of Sec. 4.2.

---

### Official Review · Reviewer_i4MY · 2022-07-12

**Rating:** 6
**Confidence:** 2
**Soundness:** 3 good
**Presentation:** 3 good
**Contribution:** 3 good

**Summary:**

This work constructs a modeling framework that aims to solve various problems in the circuit design process. This work incorporates
1. A novel circuit graph that is able to jointly integrate the topological and geometrical information and is claimed to be the first unified circuit representation approach that can be easily compatible across EDA tasks and stage.
2. A novel message-passing paradigm, CircuitGNN, that is tailored towards the aforementioned graph dataset structure. The structure can conduct message-passing on both topological and geometrical edges distinctively and then fuse the messages to update cells and nets representations
3. Extensive experiments validates the merits of the proposed methods in terms of both the task accuracy and execution speeds.

**Questions:**

1. Would be great to provide more summarized details on the proposed models and the baselines in terms of the number of parameters and operations and their types.

**Limitations:**

Yes, it's addressed.

**Strengths And Weaknesses:**

Strength:
1. This work does a good job on analyzing and illustrating the tasks and problems of circuit EDA in light of the machine learning methods.
2. The methodology is described in much detailed but straight-forward way.
3. Overall this work provides decent improvements over the existing methods. Just per the results alone, it is impressive.
4. The code provided in the supplementary materials is certainly a plus, contributing to the transparency and reproduction of the works in the fields.

Weakness:
1. Apart from the improvements on the message passing methods, one of the key contributions of this work is to be able to jointly integrate the topo and geom information in one model. However, I do not see clearly the motivation for this point from both the application and results perspectives. For actual application, is there a significant disadvantage of simply using two sets of models or even methods respectively for logical synthesis and place-and-routing? One of the reasons, I would perceive, is that a joint model may yield a better task performance due to the complementary information. However, as Table 2 suggests, the joint model's improvements against the proposed method with only geom message passing.

---

> ### Author Response · Authors · 2022-07-31
> **Official Reply to Reviewer i4MY**
>
> Thank you for your inspiring comments. Here are some points we would like to clarify:
>
> ## About the weakness you mentioned
>
> 1. The necessity of topology and geometry’s cooperation is demonstrated both methodologically and experimentally:
>     1. Methodologically, topology and geometry jointly decide most circuit properties e.g. congestion and net length mentioned in this paper. Circuit congestion is mainly caused by crowdedly-placed cells (geometry) and dense nets (topology). Net length is determined by cells’ connectivity via nets (topology) and the distances among connected cells (geometry).
>     2. Experimentally, take Congestion Prediction as an example: (**N** refers to Node-level and **G** refers to Grid-level)
>
>
>         |  | time | pearson (N) | spearman (N) | kendall (N) | pearson (G) | spearman (G) | kendall (G) |
>         | --- | --- | --- | --- | --- | --- | --- | --- |
>         | Ours (w/o. geom.) | 21.62 | 0.779 | 0.289 | 0.217 | 0.315 | 0.468 | 0.329 |
>         | Ours (w/o. topo.) | 21.54 | 0.883 | 0.713 | 0.573 | 0.684 | 0.730 | 0.536 |
>         | Ours | 27.07 | **0.887** | **0.714** | **0.575** | **0.697** | **0.770** | **0.577** |
>
>         Although Congestion Prediction in the placement stage result mainly depends on the geometry, integrating both topology and geometry can yield even better performance.
>
>
>     Therefore, a joint model is needed to capture deep interaction between topology and geometry and generate informative representations for downstream tasks.
>
>
> ## About your q**uestions:**
>
> 1. Some details of our model and baselines are listed below:
>
> Congestion Prediction
>
> | Model | # parameter | Inference Time (s) | Topology-aware | Geometry-aware | Compatiblity to logic/placement |
> | --- | --- | --- | --- | --- | --- |
> | GCN | 205K | 2.74 | yes | no | logic |
> | GraphSAGE | 204K | 2.69 | yes | no | logic |
> | GAT | 205K | 3.18 | yes | no | logic |
> | CongestionNet | 280K | 2.99 | yes | no | logic |
> | pix2pix | 992K | 0.35 | no | yes | placement |
> | LHNN | 54K | 65.21 | yes | yes | placement |
> | Circuit GNN | 480K | 4.09 | yes | yes | both |
>
> Net Length Prediction
>
> | Model | # parameter | Inference Time (s) | Topology-aware | Geometry-aware | Compatiblity to logic/placement |
> | --- | --- | --- | --- | --- | --- |
> | MLP | 4K | 0.60 | no | no | neither |
> | Net^2f^ | 13K | 1.13 | yes | no | logic |
> | Net^2a^ | 39K | 2.39 | yes | no | logic |
> | LHNN | 54K | 21.41 | yes | yes | placement |
> | Circuit GNN | 694K | 3.51 | yes | yes | both |
>
> Note that **Inference Time** is evaluated on *superblue19*[1] (with 496045 cells, 515951 nets and 1912420 pins).
>
> [1] Bustany et al. ISPD 2015 Benchmarks with Fence Regions and Routing Blockages for Detailed-Routing-Driven Placement.

---

### Official Review · Reviewer_vsvz · 2022-07-16

**Rating:** 6
**Confidence:** 3
**Soundness:** 3 good
**Presentation:** 3 good
**Contribution:** 3 good

**Summary:**

This paper proposed a novel graph representation: Circuit Graph, integrating the heterogeneous circuit information from logic synthesis and placement to facilitate the EDA design process. The proposed graph structure considers both topological (cell connection in the netlist) and geometric information (positioning of the standard cells on the layout). A corresponding graph neural network (GNN) structure is proposed for extracting circuit representation for various downstream tasks. The experimental results demonstrated the effectiveness of the graph in congestion and net wirelength prediction tasks with efficient NN computation.

**Questions:**

1. The reviewer wonders if the authors can make comments about how transferrable the proposed GNN features are.
1. The authors claimed the proposed GNN is very efficient in terms of computation. To be used in practical cases, the time consumed in training and inference of GNN should be compared with the traditional EDA routing tool. Some simple statistics over this can help justify the actual impact of the usage in EDA.
1. Detailed GNN formulation question 1): In equation 5, a matrix $W_{\varepsilon^T\rightarrow\mathcal{U}}$ is used in topological message passing, and in equation 7, a vector $a$ is used in the geometric message passing. What is the consideration underneath the difference here?
1. Detailed GNN formulation question 2): To fuse the topological and geometrical messages, maxpooling is used (equation 3). What is the underlying thought process to use maxpooling for fusion, and what may be the effect of other fusion methods (e.g. adding, concatenation, averagepooling)?
1. Detailed GNN formulation question 3): The readout for cell and net also include the raw features. The reviewer wonders how important those raw features are. If excluding them from the readout results in significant performance drops, the usefulness of the extracted GNN representation should be questioned.

**Limitations:**

The paper mentioned that one limitation is to test the proposed method under commercial products and more complex scenarios. The reviewer appreciate that the authors bring up this and understand the difficulty behind it.

**Strengths And Weaknesses:**

Strengths:
1. Heterogeneous information fusion across multiple EDA design stages. Typically, circuit designs are divided into multiple phases. Each phase may have its own unique representation for the same underlying circuit. The proposed circuit graph brings two representations (netlist and cell placement) into a unified graph representation, which provided a more informative data structure embedding knowledge from multiple EDA design phases.
1. The proposed circuit graph is general enough to be extended to inspire future work. The paper only touches on congestion and net wirelength prediction tasks for detailed routing, and the graph featurization contains only related basic topology information and simple geometric information. The reviewer believes the proposed graph can inspire more work in EDA areas. For example, by adding standard cell delay as one new feature in the cell node, the proposed graph may also help with the timing analysis of the circuit.
1. The overall GNN structure follows the design of the circuit graph, which sounds promising. The topological and geometric message passing structures preserve the structure of the original circuit graph,

Weaknesses:
1. The paper didn't touch on how representative the extracted GNN features are. The two tasks (congestion prediction and net wirelength prediction) in the paper are experimented independently. Although these two tasks have different readouts, they shared the same input graph features and extract GNN feature representation. It would be interesting to check if the knowledge can be transferred from one task to another using the proposed GNN.
1. Although the overall GNN structure sounds promising, some detailed formulation or design choice of GNN needs to be further justified. Detailed comments are made in the questions.

---

> ### Author Response · Authors · 2022-07-31
> **Official Reply to Reviewer vsvz PART 2/2**
>
> ## About your questions 3-5
>
> 3. On the one hand, there are usually more geom-edges than topo-edges in Circuit Graph (3.4M geom-edges and 1.9M topo-edges in *superblue19*[2]), so we use edge-weight summation rather than inner product, which is $F_{\cal U}$ (hidden dimension of net) times more expansive in computation. On the other hand, it is also reasonable for geom-edges to use edge-weight summation because geometrically closer cells have a stronger relationship. Still, we test the performance when topo-edges use edge-weight summation or geom-edges use inner product: (**N** refers to Node-level and **G** refers to Grid-level)
>
>
>     |  | time | pearson (N) | spearman (N) | kendall (N) | pearson (G) | spearman (G) | kendall (G) |
>     | --- | --- | --- | --- | --- | --- | --- | --- |
>     | topo w. edge-weight summation | 25.35 | 0.886 | 0.707 | 0.570 | 0.694 | 0.743 | 0.552 |
>     | geom w. inner product | 37.71 | 0.886 | **0.717** | **0.579** | 0.689 | 0.734 | 0.542 |
>     | Ours | 27.07 | **0.887** | 0.714 | 0.575 | **0.697** | **0.770** | **0.577** |
>
>     We can see that:
>
>     - If we use edge-weight summation to process topo-edges, time is marginally saved but performance gets a bit worse.
>     - If we use the inner product to process geom-edges, time cost will increase with no significant improvement.
> 4. We hope to keep most of the informative values when fusing the topological and geometrical information, while sum-pooling and mean-pooling may revise them. Concatenation is not considered because we hope to keep the same hidden dimension in each layer. The results below show that using sum-pooling and mean-pooling has worse spearman (G) & kendall (G) and only marginal improvement in other metrics: (**N** refers to Node-level and **G** refers to Grid-level)
>
>
>     |  | time | pearson (N) | spearman (N) | kendall (N) | pearson (G) | spearman (G) | kendall (G) |
>     | --- | --- | --- | --- | --- | --- | --- | --- |
>     | Ours (sum pool) | 29.00 | 0.887 | **0.717** | **0.580** | **0.699** | 0.756 | 0.564 |
>     | Ours (mean pool) | 29.38 | **0.888** | 0.715 | 0.577 | 0.697 | 0.755 | 0.563 |
>     | Ours | 27.07 | 0.887 | 0.714 | 0.575 | 0.697 | **0.770** | **0.577** |
> 5. We concatenate the raw features to enrich the representations. The results below show that excluding raw features only causes a marginal performance drop: (**N** refers to Node-level and **G** refers to Grid-level)
>
>
>     |  | time | pearson (N) | spearman (N) | kendall (N) | pearson (G) | spearman (G) | kendall (G) |
>     | --- | --- | --- | --- | --- | --- | --- | --- |
>     | Ours (w/o. raw feat.) | 27.45 | **0.892** | 0.713 | 0.574 | **0.697** | 0.759 | 0.567 |
>     | Ours | 27.07 | 0.887 | **0.714** | **0.575** | **0.697** | **0.770** | **0.577** |
>
> [1] Bustany et al. ISPD 2015 Benchmarks with Fence Regions and Routing Blockages for Detailed-Routing-Driven Placement.
>
> [2] Dai et al. NCTU-GR: Efficient Simulated Evolution-Based Rerouting and Congestion-Relaxed Layer Assignment on 3-D Global Routing. In TVLSI 2010

---

> ### Author Response · Authors · 2022-07-31
> **Official Reply to Reviewer vsvz PART 1/2**
>
> We appreciate your constructive review and valuable questions. Here are some points we would like to clarify:
>
> ## About your questions 1-2
>
> 1. Our motivation is to propose a versatile model to handle diverse EDA tasks on multi-stage circuits without repeatedly designing individual models:
>     1. In Circuit Graph, we use topo-edges and geom-edges to capture the topology and geometry, which determine most of the properties of a circuit.
>     2. In Circuit GNN, to capture the deep topological and geometrical relationships and generate informative representations for diverse EDA tasks, we pass the messages through topo-edges and geom-edges and fuse them at the end of each layer.
>
>     The experimental results in Sec. 5 show that our model can generate informative representations with potential transferability to handle diverse EDA tasks.
>
>     It is a good idea to further evaluate the representativeness of extracted GNN features by conducting a transfer experiment. Following your suggestion, we train Circuit GNN/LHNN with Congestion Prediction task and evaluate/fine-tune them with Net Length Prediction:
>
>     Some experiment settings:
>
>     - For the **evaluation setting**, a different readout module is trained from scratch, but GNN’s parameters are fixed.
>     - For the **fine-tuning setting**, a different readout module is trained from scratch, and GNN’s parameters are fine-tuned at the same time.
>     - For **evaluation & fine-tuning**, LHNN and Ours are only trained for 1/5 epochs of default setting, to show the transferability of learned features.
>
>     |  | time | pearson | spearman | kendall |
>     | --- | --- | --- | --- | --- |
>     | MLP | 2.22 | 0.493 | 0.547 | 0.415 |
>     | LHNN (evaluate) | 192.45 | 0.689 | 0.715 | 0.563 |
>     | Ours (evaluate) | 9.55 | **0.799** | **0.811** | **0.622** |
>     | LHNN (fine-tune) | 248.96 | 0.805 | 0.794 | 0.612 |
>     | Ours (fine-tune) | 14.80 | **0.842** | **0.829** | **0.639** |
>     | LHNN | 260.00 | 0.801 | 0.796 | 0.603 |
>     | Ours | 14.79 | 0.848 | 0.835 | 0.646 |
>
>     The results show that the knowledge Circuit GNN learns from Congestion Prediction can be easily transferred to another task (no matter for direct use or fine-tuning), while LHNN is weak in transferability.
>
> 2. From the traditional EDA routing tools, NCTUGR [2] is one of the most commonly used tools for estimating the congestion values given the circuit layout. To compute the congestion label for the testing circuit we used (*superblue19*), the run time is about 50s, which is considerably much higher than the inference time for our proposed circuit GNN (4.1s) as well as other learning based method in the literature.  Besides, the inference time of Circuit GNN on *superblue19*[1] (with 496045 cells, 515951 nets, and 1912420 pins) is 4.09s for Congestion Prediction and 3.51s for Net Length Prediction. It is around an order of magnitude faster than SOTA LHNN (65.21s and 21.41s).

---

### Author Response · Authors · 2022-08-09
**Update Rebuttal Version**

Thanks for all reviewers' constructive suggestions, which help us find some points we didn’t explain clearly. Now we have got them clarified in a new version. The revision mainly includes:
- A new baseline (MPNN) is added into Congestion Prediction (see **Table 1**), and the explanation of why our message function is better than MPNN in **Section 5.3**.
- More details of baselines (e.g. # of parameters and inference time, see **Table 16-19** in **Appendix F**), and the inference time of each step of DREAMPlace (see statement in **Appendix F**).
- A transfer learning task to further evaluate the representativeness of extracted GNN features of SOTA LHNN and our model (see **Appendix G**).
- Model Sensitivity Experiments (see **Appendix D**), containing:
	1. The justification of topological and geometrical message function choices;
	2. The justification of using max-pooling to fuse topological and geometrical information;
	3. The justification of concatenating raw features in readout representation;
	4. The justification of designing geom-edges instead of directly encoding cell positions.
- Update **Limitation** in **Section 6**.

---

### Public Comment · ~Shuwen_Yang1 · 2023-04-15
**Code Available**

https://github.com/PKUterran/NetlistGNN

---

### Meta-Review · Area_Chair_D1yk · 2022-08-31

**Recommendation:** Accept
**Confidence:** Certain

**Metareview:**

This paper proposes a GNN approach to EDA using the construction of a circuit graph that combines geometric and topological information, as well as features generated from physical properties of circuit components. While reviewers have raised certain concerns (some addressed already in rebuttal), they all settled (post rebuttal) on recommending weak accept of the paper. I agree with them and think the NeurIPS audience would benefit from the inclusion of this work in the program, and therefore I recommend acceptance. I would like to encourage the authors to take into account the comments and discussion with the reviewers, as well as incorporate materials presented in their responses, when preparing the camera ready version.

**Award:**

No

---

### Decision · Program_Chairs · 2022-09-14

Accept